# Phenotypic and Genetic Components for Growth, Morphology, and Flesh-Quality Traits of Meagre (*Argyrosomus regius*) Reared in Tank and Sea Cage

**DOI:** 10.3390/ani11113285

**Published:** 2021-11-17

**Authors:** Antonio Vallecillos, Emilio María-Dolores, Javier Villa, Francisco Miguel Rueda, José Carrillo, Guillermo Ramis, Mohamed Soula, Juan Manuel Afonso, Eva Armero

**Affiliations:** 1Department of Agronomic Engineering, Technical University of Cartagena, Paseo Alfonso XIII 48, 30202 Cartagena, Spain; antonio.vallecillos@edu.upct.es (A.V.); emilio.mdolores@upct.es (E.M.-D.); 2Alevines del Sureste S.L., calle Cabo Cope s/n, Águilas, 30880 Murcia, Spain; j.villa@avramar.eu (J.V.); f.rueda@avramar.eu (F.M.R.); j.carrillo@avramar.eu (J.C.); 3Department of Animal Production, University of Murcia, Avenida Teniente Flomesta 5, 30100 Murcia, Spain; guiramis@um.es; 4Institute of Sustainable Aquaculture and Marine Ecosystems (GIA-ECOAQUA), Carretera de Taliarte s/n, Telde, 35214 Las Palmas, Spain; mohamed@anfaco.es (M.S.); juanmanuel.afonso@ulpgc.es (J.M.A.)

**Keywords:** meagre (*Argyrosomus regius*), infrared spectroscopy (NIR), moisture, fat content, protein, collagen, heritabilities, correlations, stock density

## Abstract

**Simple Summary:**

Meagre is an emergent species in aquaculture, due to its fast growth rate, low feed conversion ratio, and the high quality of the product. Although advances have been achieved in its management, reproduction, and feeding, breeding programs have not yet been developed. For this reason, this study aimed to provide information about the genetic variations in growth, morphology traits, and flesh chemical composition to be included in a selective breeding program, studied in two different housing systems (cage and tank). Heritabilities for growth and morphology traits, and for fillet fat percentage were medium, revealing those traits as a possible selection criterion in a breeding program. Image analysis provided a great amount of objective information regarding the different morphological traits of the fish, where a positive and high correlation with growth traits was observed. Positive phenotypic correlation between fillet fat percentage and body weight was observed, so a selection process to improve growth rate could lead to a fish with higher fillet fat percentage.

**Abstract:**

Meagre (*Argyrosomus regius*) plays an important role in the aquaculture system, with the potential for diversification of European aquaculture, and is characterized by its fast growth rate, low feed conversion ratio, and the high quality of the product. Focusing on the relevance of meagre, the aim of the study was to analyze growth performance, fish morphology, and flesh composition phenotypically and genetically to be considered as a strategy in a breeding program. For this purpose, 633 fish were raised in two different housing systems, in sea cages or in a continental tank, and when they reached harvest size, manual growth traits, automatic morphology by the image analysis program IMAFISH_ML, and flesh chemical composition (fat, protein, moisture, and collagen percentages) were measured. The fish reared in the cages showed a higher body weight and fillet fat percentage than those in the tank. Heritabilities for growth and morphology traits, and for fillet fat percentage were medium, revealing these traits as a possible selection criterion in a breeding program. Phenotypic and genetic correlations between growth and morphology traits were positive and high. Phenotypic correlations between growth or morphology traits with fillet fat percentage were positive and medium; genetic correlations were not estimated accurately.

## 1. Introduction

The meagre (*Argyrosomus regius*) is one of the most important fish in the diversification of Mediterranean aquaculture [1]. It is a teleost fish of the Sciaenidae family which can be found along the Atlantic coast of Europe, in the Mediterranean Sea and Black Sea, and the east coast of Africa, at depths of between 15 and 200 m [2]. It is also found in estuaries and coastal lakes [3]. Meagre has an important role in the aquaculture system, as a potential species for diversification of European aquaculture, with its annual production in the Mediterranean area exceeding 41,000 tons [4]. This species is easy to adapt to captivity due to its tolerance to a wide range of salinity (5–39%) and temperature (13–28 °C) [2]. The main interest in this species lies in its fast growth rate (around 800–1000 grams in 18 months) with a low feed conversion ratio [5]. In addition, meagre have an attractive body shape as a whole fish commodity, a low fat fillet content, and a good processing yield [6]. Moreover, meagre has been proven, along with other aquaculture species, to be a great source of unsaturated fatty acids [7,8,9], which are highly appreciated in human nutrition for their great benefits in several cardiovascular diseases [10,11].

Strategies need to be established in the different areas of knowledge to promote the consolidation of the meagre industry. Companies and research already support different actions in the field of feeding, with studies that have tested different diets, with different protein percentages or including new feeds [5,7,12,13,14], or reproduction and batch management, in which different reproduction techniques have been studied and how stocking density affects the final product [7,13,15,16,17,18,19]. Most studies into disease prevention have focused on granulomas in all soft tissues [20,21,22,23].

However, strategies in aquaculture breeding programs are scarce, and particularly in an emerging species such as the meagre. Selective breeding programs are one of the fundamental strategies that must be included in any production system. In the case of terrestrial livestock, the accumulative annual genetic gain rate is estimated to be around 1–3% [24]. The different programs in Europe are currently considering different objectives, such as growth performance and feed efficiency, morphology, product quality, processing yield, reproduction, and disease resistance [25]. The selection criterion greatly affects the genetic progress through differences in additive genetic variation, which is essential for improving breeding values [26]. For that reason, it is important to know about the genetic variation for the traits of interest in fish farming.

Thus, growth and feed conversion stand out as one of the main objectives in breeding programs, due to the high feeding costs. Due to the difficulty in measuring feed consumption, feed conversion is normally improved indirectly through growth traits such as weight and length [27].

Regarding fish quality, the external appearance is one of the most important traits as irreversible modifications of morphology produce alterations with respect to the quality standard. In this case, several deformities have been described in meagre, such as granulomas [20]; this can lead to reduced growth performance and fish that are visually unacceptable for the market.

To assess external morphology, Navarro et al. [28] developed a non-invasive methodology that consists of a fast and automated software for image analysis (IMAFISH_ML), measuring 27 fish morphometric traits (technological traits) on three commercial fish species—gilthead seabream (*Sparus aurata* L.), meagre (*Argyrosomus regius*), and red porgy (*Pagrus pagrus*). Methodological tools, based on image analysis, have also been developed for the on-line sorting of farmed seabass (*Dicentrarchus labrax*, L.) by size, sex, and the presence of abnormalities [29].

To the best of our knowledge, only one work [30] has studied genetic variation for growth traits in meagre, albeit only for body weight and length, whilst no study has considered the genetic component for other traits related to external morphology.

In other aquaculture species, other breeding objectives such as flesh quality and disease resistance have been established later [25,31,32,33,34]. In meagre, there are no such studies of genetic variation for flesh composition and disease resistance, although the expression of the immune system genes has been analyzed [35].

The aim of the present study was to estimate genetic parameters of the main traits that are of fish farming interest, such as growth performance, morphology, and flesh composition, for the first time in meagre (*Argyrosomus regius*), to be considered as a strategy in a selective breeding program.

## 2. Materials and Methods

### 2.1. Ethics Statement

To ensure that animal welfare standards were maintained, anesthetic was used in the sampling procedure. All animal experiments described in this manuscript fully complied with the recommendations in the Guide for Care and Use of Laboratory Animals of the European Union Council Universidad Politécnica de Cartagena of Región de Murcia, Spain (approval number CEI21_006).

### 2.2. Animals

The experiment was carried out with meagre, which were obtained from one broodstock (n = 9; 4 males and 5 females), belonging to the company Alevines del Sureste S.L. of the Avramar group. The broodstock were under a controlled photoperiod (8L:16D) to synchronize maturation, all broodstock were injected with LH-RH to induce spawning in April 2019, egg release was initiated in early May 2019. During that period, the animals were fed ad libitum with Vitalis Repro (Skretting, Stavanger, Norway), and egg production was monitored daily. When the total egg production stabilized, the egg batch was established at the beginning of May 2019. Eggs from the broodstock were collected and pooled for four consecutive days (4DL model) to maximize family representation. In total, 1.5–2 egg kg were incubated in cylinder conical tanks (650 L). Water conditions were as follows: temperature 20.0 °C, salinity 36%, and oxygen saturation was 100–120%. Thus, 633 offspring were individually tagged in the abdominal cavity for individual identification at 220 days post-hatching (dph), with a Passive Integrated Transporter (PIT, Trovan Daimler-Benz, UK), following the tagging protocol described by Navarro et al. (2006) [36], and a sample of caudal fin was collected and preserved in absolute ethanol at room temperature for future DNA extraction. Thirty days later, fish were moved to different facilities of the company Avramar S.L. where they were reared in two different housing systems (HS): (1) 255 fish were allocated in a cage in the Mediterranean Sea (Villajoyosa, Alicante, Spain) under intensive conditions. The cage of 12 meters in diameter was anchored at a depth of 38 meters with a stock density of 10.6 kg/m^3^ and water temperature of 20.4 °C (13–28 °C), salinity 34%, and oxygen saturation was 90%; (2) 378 fish were in a continental tank under intensive conditions at the Avramar S.L. facilities (Cabo Cope Aguilas, Murcia, Spain) with a stock density of 15 kg/m^3^. The rectangular tank had a capacity of 20 m^3^, with water temperature of 21.0 °C (19–23 °C), salinity 36%, and oxygen saturation was 100%. At the end of the trial, a total of 633 fish reached harvest size (549 dph) and fish were slaughtered by immersion in ice cold water (hypothermia) and stored at 4 °C for 24 h for further processing.

### 2.3. Microsatellite Genotyping and Parental Assignment

DNA was extracted from the caudal fin using the DNA kit (E.Z.N.A.^®^ Tissue, Norcross, GA, USA), and then kept at 4 °C. Subsequently, DNA quantity and quality were determined with a NanoDrop™ 2000 spectrophotometer v.3.7 (Thermo Fisher Scientific, Wilmington, NC, USA). All the fish—broodstock and offspring—were genotyped for 10 microsatellite loci multiplex (Table 1), using the Type-it Microsatellites PCR kit (QIAGEN^®^, Hilden, Germany).

The PCR was performed in a 12.5 μL reaction mix with a concentration of 10 μmol/L for each primer and 10 ng/μL template DNA. The thermal profile included a pre-denaturation step of 95 °C for 15 min followed by 32 cycles of denaturation–annealing–extension at 94 °C for 30 s, 57 °C for 90 s, and 72 °C for 1 min and one final elongation step at 60 °C for 30 min. Amplicons were resolved by capillary electrophoresis on a 3500 Genetic Analyzer (Applied Biosystem, Foster City, CA, USA) using LIZ500 size standard marker. The fragment size was analyzed using Microsatellite analysis cloud (Thermo Fisher Scientific, Waltham, MA, USA), which was used for genotyping. For the parental assignment the exclusion method as implemented in VITASSING (v.8_2.1) software [37] was used, where the assignment of parents, with an assignment error in the batch gate of 3.9% and batch tank of 1.8%. Finally, 616 fish were used in the different analyses, 245 from the cage and 371 from the tank.

### 2.4. Measurements

#### 2.4.1. Manual Growth (MG) Measurements

Body weight (BW) was measured with scales accurate to 1 g, and total length (TL) was measured with an ichthyometer with 1 mm divisions.

#### 2.4.2. Automatic Morphology (AM) Measurements

Immediately afterwards, each fish was photographed with a digital camera, side view, in a dark box with controlled light as per Navarro et al (2016) [28]. Later, all pictures were analyzed, in a standardized way, by applying the automatic image analysis IMAFISH_ML software, developed in MatLab v.7.5 (MathWorks, Natick, MA, USA) [28]. All defined traits are shown in Table 2, and a fish picture indicating length and height measurement is depicted in Figure 1.

#### 2.4.3. Flesh Chemical Composition

The fish were then manually eviscerated, skinned, and filleted, with the fillets being frozen at −20 °C for further analysis of the flesh chemical composition.

For flesh chemical composition, the fillets were homogenized through a mixer, and the percentages of protein, fat, moisture, and collagen were estimated by the indirect method of near-infrared spectroscopy (near infrared spectroscopy, NIR), using FOODSCAN LAB equipment (FOSS IBERIA, Barcelona, Spain).

### 2.5. Statistical Data Analyses

For phenotypical analysis:

Numerical data for each trait were tested for normality and homogeneity of variances using SPSS^®^ (v.26.0) [43] and were analyzed with two general linear models (GLMs):Y_ij_ = µ + HS_i_ + e_ij_; for MG and AM measurements 
Y_ij_ = µ + HS_i_ + b ∗ BW_j_ + e_ij_; for flesh composition (1)
where Y_ij_ is an observation of an individual j in the housing system (HS) i, μ is the overall mean, HS is the effect of the environmental conditions because of the HS (I = cage or tank), b is the regression coefficient between the analyzed variable and the covariate BW, and e_ij_ is a random residual error.

For genetic parameter estimates:

For each combination of two traits, genetic parameters were estimated under a Bayesian approach using a two-trait animal mixed model which can be written as:(2)y=Xβ+Zu+Wp+e
where **y** is the vector of data; **β** is the vector of systematic effects including the HS (2 levels: cage or tank) for all traits and the covariate body weight only for flesh composition traits; **u** is the vector of additive genetic effects; **p** is the permanent environmental effect of family-HS; **e** is the residual; and **X**, **Z**, and **W** are incidence matrices relating data with systematic effects and random additive genetic and permanent effects, respectively.

The systematic effects, **β** were assumed a priori to follow uniform distributions. The a priori distribution of the additive genetic effect was p(a|G)~N(0, G⊗A), where G is the 2 × 2 additive genetic covariance matrix between traits, A is the numerator relationship matrix, of dimension N, equal to the number of individuals in the pedigree, and ⊗ is the Kronecker product. The a priori distribution of permanent environmental effects was $p(p|P)\sim N\left(0,\;P⊗I_{p}\right)$, where **P** is the 2 × 2 covariance matrix of permanent environmental effects between traits and $I_{p}$ is the identity matrix. Similarly, the distribution of the residual effects was $p\left(e|R\right)\sim N\left(0,R⊗I_{e}\right)$, where **R** is the corresponding 2 × 2 residual covariance matrix between traits and $I_{e}$ is the identity matrix. Bounded uniform priors were assumed for the elements of **G**, **P**, and **R**.
(3)$$G=\left[\begin{array}{cc} \sigma_{u1} & \sigma_{u1,u2}\\ \sigma_{u2,u1} & \sigma_{u2} \end{array}\right],P=\left[\begin{array}{cc} \sigma_{p1} & \sigma_{p1,p2}\\ \sigma_{p2,p1} & \sigma_{p2} \end{array}\right],\;R=\left[\begin{array}{cc} \sigma_{e1} & \sigma_{e1,e2}\\ \sigma_{e2,e1} & \sigma_{e2} \end{array}\right]$$
the marginal posterior distributions of all the unknowns were approximated by Gibbs sampling using the gibbs3f90 software [44]. Sampling processes of 200,000 iterations each were run. The first 50,000 iterations were discarded as burn-in and samples of the parameters of interest were saved every five iterations. The sampling variance of the chains was obtained by computing Monte Carlo standard errors. Statistics for the marginal posterior distributions were calculated directly from the samples using the R package “BOA” (R Development Core Team, 2016) [45]. The magnitude of estimated heritability was established following the classification recommended by Cardellino and Rovira (1987) [46], as low (0.05–0.15), medium (0.20–0.40), high (0.45–0.60), and very high (>0.65). The magnitude of correlation was established following the classification of Navarro et al. (2009) [47], as low (0–0.40), medium (0.45–0.55), and high (0.60–1), regardless of whether they were positive or negative.

## 3. Results

### 3.1. Phenotyping

The phenotyping results for MG (BW and TL) and AM measurements (SL, CPH, and FHC) are shown in Table 3. The fish reared in the cage showed higher BW (38% heavier than the average), TL, SL, CPH, and FHC (15, 16, 15, and 18% greater than the average, respectively), than those in the tank. The much higher increase in BW than length or height indicated that the growth was more in volume than in the area for this weight range.

To know if the meagre’s shape is affected by BW, the SL/FHC ratio was calculated including BW as covariate (Table 3). Therefore, we observed that when BW was adjusted to average BW equal to 995 g, there was no difference between fish reared in the cage or in the tank. However, BW had a significant negative effect on the SL/FHC ratio although it was very reduced, thus when fish increased their weight by 100 g this ratio decreased 1.61 × 10^−2^ (footnote Table 3); increased BW in meagre involved slight changes in shape, gaining more in height than in length. For the harvest BW, the phenotypical variation for this ratio ranged from a minimum and a maximum value of 3.11 to 4.79, with 95% of the fish being in the interval [3.93–4.05], so the meagre were four times longer than they were wide.

The flesh composition (moisture, protein, fat, and collagen percentages) is shown in Table 4. The BW showed a negative effect on moisture, and positive on the fat percentage; thus when the fish weight increased 100 g, the moisture decreased 0.3% and the fat increased 0.2%. When flesh composition was adjusted to an average BW of 995 g, the cage fish showed a higher fat percentage (33% better than the average) and lower protein percentage (−8.7% better than the average) than the tank fish.

### 3.2. Microsatellite Genotyping and Parental Assignment

For the cage fish, 97.2% of the offspring were assigned, of which 94.5% were assigned to a single couple of parents. In the case of the tank fish, a 98.9% parental assignment was obtained, of which 97.9% were assigned to a single couple. After the assignment, unequal breeder contribution was observed. In the cage fish, one out of five females produced 45.3% of the offspring although all the females contributed to the offspring, and one out of four males contributed with 59.5% of the offspring and all the males contributed. Similarly, in the tank fish, two out of five females contributed with 55.5% of the offspring and all females contributed, whilst two out of four males contributed with 63.8% of the offspring and all the males contributed. Pedigree construction using selected highly informative microsatellite markers yielded 20 full-sib families for the cage fish with a mean of 15.06 sibs (range 2–58 sibs), and in the tank fish, it produced 20 full-sib with a mean of 19.47 sibs (range 7–60 sibs).

### 3.3. Heritabilities and Correlations

#### 3.3.1. Heritabilities

Heritabilities estimated for each trait and genetic and phenotypic correlations between traits are shown in Table 5. In general, heritability estimates showed medium and low values with a high standard error, so the results should be viewed with caution. In the case of MG traits, the heritabilities were medium for BW and TL. For AM traits, SL and FHC showed medium heritability, and CPH and SL/FHC low. For flesh composition, fat and moisture percentages showed medium heritability, and protein and collagen low heritability.

#### 3.3.2. Correlations

Within group of traits:

The phenotypic correlations were estimated with high accuracy. They were positive and high between MG traits, BW–TL, and between AM traits (SL, FHC, and CPH), except for the SL/FHC ratio which was almost null with length measurements (TL and SL) and low and negative with height measurements (CPH and FHC). For flesh composition they were positive and low for collagen–moisture, negative and high for fat–moisture, medium for fat–protein, and negative and low for protein–collagen, protein–moisture, and collagen–fat.

The genetic correlations between the MG traits, BW–TL, were very high. Between the AM traits they were positive and high for all the traits (SL, FHC, and CPH) except for SL/FHC, which showed similar patterns but were estimated with low accuracy.

In the case of flesh composition traits, most of them were estimated with little accuracy, with a high standard error, except for the correlation between fat and moisture, and therefore safely interpreting the corresponding genetic correlations is difficult. Taking this consideration into account, the genetic correlations estimated were positive and low for moisture–collagen, negative and high for fat–moisture, negative and medium for fat–protein and fat–collagen, and negative and low for protein–collagen and moisture–protein. Genetic correlations were in the same order as the phenotypic correlations.

Between groups of traits:

Since AM traits (SL, CPH, and FHC) are another way to analyze the MG (BW and TL), the phenotypic correlations between them were positive and high in all the cases, except for the SL/FHC ratio, which were low and negative. The BW correlations were higher with height than with length measurements. Considering these results, at this weight range, the fish is growing slightly more in height than in length.

Regarding MG traits and flesh composition, phenotypic correlations were positive and medium for fat, negative and medium for moisture and low for protein, and almost null for collagen. For the AM traits and flesh composition the phenotypic correlations followed a similar pattern to the MG traits and flesh composition, apart from the SL/FHC ratio. In the case of the SL/FHC ratio, the correlations were almost null with protein and collagen, medium with fat and moisture, and negative for fat and positive for moisture.

Genetic correlations between MG (BW and TL) and AM traits (SL, CPH, and FHC) were positive and very high for all the possible combinations, except for the SL/FHC ratio, which were negative with BW and null with TL but estimated with high standard errors.

Genetic correlations between MG traits (BW and TL) and flesh composition were estimated with low accuracy and thus interpreting them safely becomes difficult. Most of these correlations were very low except for protein, with both BW and TL showing a negative and medium value.

Genetic correlations between the AM traits and flesh composition showed higher values than those with the MG traits, highlighting the negative correlations between fat and SL, CPH, and SL/FHC and, conversely, the positive correlations between moisture and SL, CPH, and SL/FHC. Protein showed negative correlations with all the AM traits and collagen showed negative correlations with SL and FHC, and positive with CPH and SL/FHC, although all of them had high standard errors.

## 4. Discussion

This study is one of the first in which genetic parameters for growth, morphological traits, and flesh composition (fat, protein, moisture, and collagen percentages) have been studied on meagre (*Argyrosomus regius*) from two batches, one reared in a cage and the other in a tank. Growth and morphological traits play a very important role in terms of economic performance in the aquaculture industry. Flesh composition, especially fat content and its fatty acid profile, is one of the characteristics that consumers appreciate most.

The first step when estimating genetic parameters was the design of the paternity. Using microsatellite markers is a widespread method for parental assignment [30,48,49,50,51]. In our case, with a 10-microsatellite panel we had a slightly higher parental assignment than Nousias et al. [30] who obtained assignments of 87.5% and 95% for two batches. A crucial factor in obtaining a high percentage of assignment in a population is that the set of selected markers must be highly polymorphic. In our study, two of them (gCT15 and CA3) were not, and they could be removed in the microsatellite markers panel. The remaining eight microsatellites proved to be enough to obtain a successful assignment.

After the construction of the pedigree, measured traits were phenotypically analyzed. For growth, our results highlighted the high growth rate of meagre, as in Fountoulaki et al. (2017) [5], thereby making it very interesting for the aquaculture industry. An important factor affecting growth rate is the stock density. Thus, we observed lower BW for fish in the tank than in the cage, which is probably mostly due to the higher stock density in the tank. Piccolo et al. [13] observed a BW of about 830 g and a TL of 42 cm in meagre raised in cages with a similar stock density (9.8 kg/m^3^) to that in our work for 15 months, and Poli et al. [7] with fish reared in tanks with high stock density about 44 kg/m^3^, found that the BW was 935.5 g, 1199.6 g, and 1502.5 g when they were 24, 26, and 30 months old, respectively. Breeders are usually kept in tanks, therefore the environmental conditions in the tank are important to enable the breeders to express their best performance.

Regarding the morphological traits measured in this study, this is the first attempt to describe the shape in meagre and relate it to growth traits. Elalfy et al. [52] also used IMAFISH_ML software as a tool in breeding programs to obtain a detailed description about the fish morphology in gilthead seabream. Since image analysis is a non-invasive measurement of individuals, it can help us to select a fish as a future breeder by its own data and not through its offspring, which is more expensive and involves a longer process. In addition, the image provides us with a larger amount of objective information than can be gained from manual measurements because it does not depend on a person.

For fillet composition of commercially sized meagre, the fillet fat percentage in previous studies was slightly lower than in our work, ranging between 0.6–1% [5,53] and 3% [13] depending on the diet’s fat content. Protein percentage changed very little, and was usually 20–21% [5,13,53] depending on the diet. In our work, the cage fish showed a higher fat percentage and a lower protein percentage (18.5%) in comparison with the tank fish and with other works. The higher fat percentage in the cage fish is likely due to a lower temperature that increased appetite and energy reserves. In addition, a pronounced seasonality has been observed in fillet fat content, which reached a maximum with the replenishment of body fat stores in early autumn [54] when our fish were slaughtered. Additionally, the methodology to measure fillet fat content in our work (NIR) differed from that used in other studies [55]. Furthermore, the fillet fat percentage showed a positive phenotypic correlation with BW in accordance with Fountoulaki et al. [5]. In gilthead seabream, Elalfy et al. [52] observed an effect of the HS on protein percentage, thus protein was 19.38% when fish were cage-reared and 20.84% when estuary-reared.

In the end, the genetic component in the variation of the measured traits was analyzed. For MG and AM traits, medium heritabilities were observed for all of them except for CPH and the SL/FHC ratio, which were low. In meagre, Nousias et al. [30] observed very high heritability for BW (0.62) and TL (0.64) in populations without previous selection and under industrial farming conditions, as in our study. There are no further references about additive genetic variation in meagre. In other marine fish, Vallecillos et al. [32] obtained medium heritabilities for body weight (0.20) in juvenile gilthead seabream; similar values were obtained for BW heritability in several studies of different species [50,56,57,58].

Genetic variation has not been studied previously for AM traits in meagre (SL, CPH, and FHC). Elalfy et al. [52] obtained a medium heritability for the same variables but in gilthead seabream. In our case, we obtained a medium heritability for SL and FHC variables but the heritability for CPH was low. Moreover, the SL, CPH, and FHC measurements were closely related to the MG traits phenotypically and genetically, highlighting that they could replace MG measurements as a selection criterion in a breeding program.

Flesh composition traits, together with other quality parameters of the fish, are becoming more relevant in genetic breeding programs [25]. This is reinforced by the consumer, since every day they are more concerned about the quality of the fish farming products usually being comparable to wild ones. In this sense, it is important to know about the genetic variation of these traits. In our study, fillet fat and moisture percentages showed medium heritability, and protein and collagen percentages were low. To the best of our knowledge there has been no work about fillet composition heritabilities in meagre, although similar results have been observed in other species [50,52,59,60].

Genetic correlations for MG traits (BW and TL) were high and positive in accordance with Nousias et al. [30]. There has been no work about AM traits in meagre, but Elalfy et al. [52] studied them in gilthead seabream and observed high and positive correlations between MG and AM traits, as happened in our work. Therefore, in a breeding program improving one of them, the rest of the traits will improve indirectly. However, the phenotypic and genetic correlation was negative for the SL/FHC ratio, thus this ratio could be studied as it changes with the selection process.

Genetic correlations between the MG or AM traits and flesh composition were not estimated accurately, due to the limited amount of data. The most interesting genetic correlations are between the fillet fat percentage and the MG and AM traits. In gilthead seabream, Elalfy et al. [52] and Lee_Montero et al. [61] observed a positive and medium correlation between growth and fillet fat percentage, suggesting that selection to improve growth traits could lead to fish with a higher fillet fat percentage. Further studies are needed to improve estimates for genetic correlations and to continue with the analysis of other carcass and meat quality traits.

## 5. Conclusions

Attention should be paid to the housing conditions for raising the fish, as the fish in the tank showed worse results for growth, and breeders are typically kept in tanks. Medium heritabilities for growth, morphological traits, and fillet fat percentage revealed them as a possible criterion to be included in a breeding program. Image analysis to describe fish morphology could replace growth measurements in a breeding program, since the amount of information obtained is very high and objective, changes in fish morphology could be observed, and these were positive and highly correlated with growth traits. An increase in the fillet fat percentage could be expected with the selection process for fish growth due to the medium and positive phenotypic correlation between the fillet fat percentage and growth and morphology traits, although genetic correlations with flesh composition were not estimated accurately. Further studies are necessary to delve deeper into these aspects and into other quality traits.

## Figures and Tables

**Figure 1 animals-11-03285-f001:**
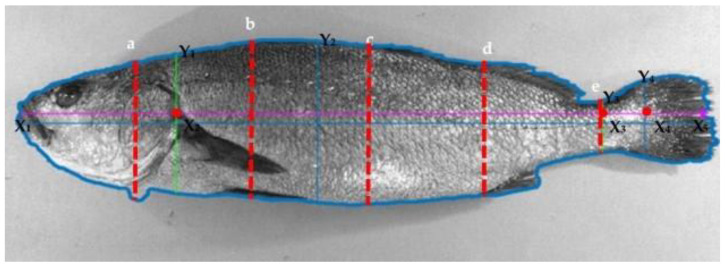
Image by IMAFISH_ML software. Lateral view, for the determination of non-invasive technological traits (NiT) of *Argyrosomus regius*: points X_1_, X_2_, X_3_, X_4_, and X_5_ of the horizontal axis are used to determine longitudinal traits; Y_1_, Y_2_, Y_3,_ and Y_4_ and a, b, c, d, and e are dorsoventral axes, which are be used to determine height traits.

**Table 1 animals-11-03285-t001:** Microsatellite panel of meagre.

Meagre STR Loci	M	F	Forward Sequence (5′-3′)	Reverse Sequence (5′-3′)	Reference
gCT15	(GCT)^7^	5′NED	ATCCGGGCGTTACTACAGTC	GTTTCTCCACACAGTGCTTTTCAGA	Porta et al. [38]
UBA50	(GT)^26^	5′NED	GCACAACTGCATCCCTTAGAT	GTTTAGAAGTGAAGACTGCGGACTG	Archangi et al. [39]
CA3	(CA)^12^	5′NED	AAGTGGAGGCTCTTACATGAAAAC	GTGACAAATTGCCTTCTGTTTCTAC	Porta et al. [38]
GA17	(GT)^12^	5′6-FAM	CTAGAGAAATTCATCCAGGGAAGTG	GTTTAGAGCAGAGAGTTAGCGGTTGTT	Porta et al. [38]
Cac mic 14	(CT)^12^	5′6-FAM	ATCTTCTCCCCTCCGTCACT	CTGTGTTGTTAAGGCGCATC	Farias et al. [40]
GA2b	(CA)^26^	5′PET	AAGTGTGGCGTCATTTCCTCT	GTATTGATGGATAGCAAGTGTCAGA	Porta et al. [38]
SOC405	(CA)^12^	5′PET	AGCCTTTTGTTTAGTTTCCCTCAT	GGGGTGTAGCAGAACCACAC	O’Malley et al. [41]
SOC431	(GT)^26^	5′VIC	GTGGTAGATGAAAACGTATAAAAGGAG	GTTTCATATATATAGTGTACAGCTCCAGCTTC	O’Malley et al. [41]
UBA53	(CA)^14^	5′VIC	TACTTCCTTCTACCCCTAAGTCTGG	GACTTTCCAGTGTAGCTGTCGTTT	Archangi et al. [39]
SOC11	(GA)^11^	5′VIC	GCCGAGTCACGAAGGAACAGAGAA	TGTCGTCTCATCTATCTCCATCTC	Saillant et al. [42]

M: motif of repetition; F: fluorescent tag.

**Table 2 animals-11-03285-t002:** The measured trait by IMAFISH_ML software.

Traits	Abbreviation	Description
Standard length (cm)	SL	Distance for X_1_–X_4_, within the horizontal axis (Figure 1).
Caudal peduncle height (cm)	CPH	Axis Y_3_ (Figure 1).
Equidistant fish height C (cm)	FHC	TLL is divided into six equal parts, then heights of each one of these five points are measured FHA, FHB, FHC, FHD, and FHE are the axes a, b, c, d, and e, respectively (Figure 1).

**Table 3 animals-11-03285-t003:** Manual growth (body weight and total length) and automatic morphology measurements (by IMAFISH_ML, software) (least square means, LSM, ±standard error, S.E.), for meagre at 549 days post-hatching raised in two different housing systems.

Housing System	Cage			Tank		
	n	LSM	S.E.	n	LSM	S.E.
BW (g)	245	1233 ^a^	18.3	371	839 ^b^	14.8
TL (cm)	245	42.7 ^a^	0.27	371	36.8 ^b^	0.22
SL (cm)	245	40.7 ^a^	0.35	371	34.7 ^b^	0.28
CPH (cm)	245	3.51 ^a^	0.03	371	3.01 ^b^	0.02
FHC (cm)	245	10.3 ^a^	0.08	371	8.59 ^b^	0.07
SL/FHC *	245	3.96	0.02	371	4.03	0.01

^a,b^ different superscripts within each row indicate significant differences between housing system (*p* < 0.05); BW = body weight; TL = total length; SL = standard length; CPH = caudal peduncle height; FHC = equidistant fish height C. * It was adjusted to BW (mean weight of 995 g) regression coefficient = −1.61 × 10^−4^ ± 3.04 × 10^−5^.

**Table 4 animals-11-03285-t004:** Flesh composition (protein, moisture, fat, and collagen percentages) (least square means, LSM, ±standard error, S.E.), for meagre at 549 days post-hatching raised in two different environments.

Housing System	Cage			Tank			Covariate BW	
	n	LSM	S.E.	n	LSM	S.E.	b	S.E.
Moisture (%)	245	74.0	0.10	371	73.9	0.08	−0.003 *	0.00
Protein (%)	245	18.5 ^a^	0.09	371	20.2 ^b^	0.07	−0.000	0.00
Fat (%)	245	5.62 ^a^	0.12	371	4.02 ^b^	0.09	0.002 *	0.00
Collagen (%)	245	1.00	0.03	371	1.00	0.02	<0.000	<0.000

^a,b^ different superscripts within each row indicate significant differences between housing system (*p* < 0.05); b = regression coefficient for moisture, protein, lipids content, and collagen were adjusted to average BW 995 g; * covariate was significant (*p* < 0.05).

**Table 5 animals-11-03285-t005:** Heritabilities (in bold at diagonal, with standard error), phenotypic correlations (below the diagonal in italics, with standard error), and genetic correlations (above the diagonal, with standard error) for manual growth, automatic morphology, and flesh composition measurements, in meagre at harvest size (549 dph).

Traits	BW	TL	SL	CPH	FHC	SL/FHC	Moisture	Protein	Fat	Collagen
BW	**0.42 ± 0.24**	0.96 ± 0.06	0.89 ± 0.19	0.90 ± 0.16	0.89 ± 0.18	−0.13 ± 0.64	0.14 ± 0.53	−0.43 ± 0.53	−0.09 ± 0.50	0.05 ± 0.58
TL	*0.91 ± 0.01*	**0.38 ± 0.22**	0.90 ± 0.16	0.88 ± 0.18	0.86 ± 0.20	−0.00 ± 0.62	0.09 ± 0.50	−0.43 ± 0.49	−0.01 ± 0.45	−0.04 ± 0.52
SL	*0.68 ± 0.04*	*0.75 ± 0.04*	**0.32 ± 0.23**	0.90 ± 0.21	0.95 ± 0.11	0.07 ± 0.71	0.29 ± 0.60	−0.22 ± 0.66	−0.37 ± 0.57	−0.34 ± 0.60
CPH	*0.66 ± 0.04*	*0.65 ± 0.05*	*0.81 ± 0.03*	**0.19 ± 0.16**	0.79 ± 0.03	0.08 ± 0.69	0.31 ± 0.57	−0.20 ± 0.65	−0.40 ± 0.53	0.32 ± 0.60
FHC	*0.74 ± 0.03*	*0.74 ± 0.04*	*0.94 ± 0.01*	*0.83 ± 0.02*	**0.39 ± 0.20**	−0.33 ± 0.65	−0.03 ± 0.66	−0.10 ± 0.68	−0.08 ± 0.63	−0.44 ± 0.57
SL/FHC	*−0.30 ± 0.10*	*−0.08 ± 0.11*	*0.03 ± 0.11*	*−0.17 ± 0.10*	*−0.31 ± 0.10*	**0.16 ± 0.15**	0.76 ± 0.37	−0.36 ± 0.64	−0.72 ± 0.39	0.26 ± 0.65
Moisture	*−0.46 ± 0.10*	*−0.41 ± 0.10*	*−0.28 ± 0.10*	*−0.31 ± 0.09*	*−0.36 ± 0.10*	*0.25 ± 0.07*	**0.32 ± 0.21**	−0.27 ± 0.62	−0.95 ± 0.07	0.08 ± 0.61
Protein	*−0.02 ± 0.12*	*0.02 ± 0.12*	*0.06 ± 0.12*	*0.09 ± 0.12*	*0.03 ± 0.12*	*0.01 ± 0.11*	*−0.11 ± 0.11*	**0.15 ± 0.14**	−0.17 ± 0.60	−0.15 ± 0.65
Fat	*0.37 ± 0.10*	*0.25 ± 0.12*	*0.17 ± 0.11*	*0.12 ± 0.10*	*0.19 ± 0.12*	*−0.21 ± 0.09*	*−0.82 ± 0.03*	*−0.38 ± 0.09*	**0.30 ± 0.20**	−0.01 ± 0.59
Collagen	*−0.01 ± 0.12*	*−0.02 ± 0.12*	*−0.04 ± 0.12*	*−0.04 ± 0.11*	*−0.04 ± 0.12*	*0.03 ± 0.10*	*0.17 ± 0.08*	*−0.24 ± 0.08*	*−0.06 ± 0.10*	**0.15 ± 0.16**

BW = body weight, g; TL = total length, cm; SL = standard length, cm; CPH = caudal peduncle height, cm; FHC = equidistant fish height C, cm; flesh composition (moisture, protein, fat, and collagen, in %). Different colors represent correlations within the same group traits (gray), correlations between AM traits and MG and flesh composition traits (yellow), and between MG and flesh composition traits (green).

## Data Availability

The data presented in this study are available on request from the corresponding author. The data are not publicly available due to privacy.

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
