# Peer review of "Phenotypic and Genetic Components for Growth, Morphology, and Flesh-Quality Traits of Meagre (*Argyrosomus regius*) Reared in Tank and Sea Cage"

_animals, 2021, doi:10.3390/ani11113285_

Round 1

Reviewer 1 Report

The manuscript by Vallecillos et al describe a study with meagre reared in two different housing systems and the phenotipic and genetic traits were determined for establishing heritability and genetic correlations. Importantly the authors established that using automatical morpholgy measurements can be used for establishing traits in a breeding program, replacing manual measurements.

The authors have used two rearing systems for growing meagre to commercial size, however the conclusions reached on comparing growth between the two systems are somehow difficult to explain. It is expected that fish reared at lower density will grow more than fish in intensive conditions, independently if they are in sea cages or land based tanks. So comparing a cage with a density that is 25% of the density on the tank is not very correct. Maybe the authors should have compared the two systems with similar density and to use experimental replicates for making a sound comparison. 

The genetic correlations presented are in line with similar works developed in other species, revealing that growth and morphology traits colud be used for selection in breeding programs, together with fat content. The fat content in meagre is known to be low and it is a quality that consumers most appreciate in this species, so the selection program should not aim at increasing fat in the fillet in a way that would have negative impacts in the consumer's perception.

As recognized by the authors, the correlations have a very high error associated and in the case of flesh composition they lack accuracy, so a study with a larger data set should be performed in order to refine the conclusions and achieve better correlations.

There are several errors in the manuscript that should be addressed, starting with the affiliations, where Juan Afonso have a mistaken affiliation. 

The english require corrections and a more clear speach. Some words appear in spanish (See figure 1 legend- Imagen....)

Mistakes or missing words were detected in the following lines:

25-26: ...process for growth fish.... should be growing fish??

52: tem, .....as a potential for diversification of European...... potential species???

115: ....fed ad libitum by Vitalis Repro... should be with Vitalis...

127-128 cage density is not intensive conditions if the value is correct

130: Oxygen saturation should be provided in percentage and not in ppm. 

Fig 1 legend: Argyrosomus regius must be in italic

The formula ? = ? + ?? + ?? + ? is missing the x

194: e is the residual ????

213-216: the references 42 and 23 should be corrected and presented in a uniform way with the remaining text

Tables 3 and 4: it appears SE or S.E. be uniform

280: medium FOR fat-protein

306: humidity???????

The bibliographic references should be carefully revised. e.g. see error in authors of Ref 19 

Author Response

Cover letter of the manuscript entitled “Phenotypic and genetic components for growth, morphology and flesh quality traits of meagre (Argyrosomus regius) reared in tank and sea cage.”

In the revised manuscript, the comments made by the reviewers have been considered and the main changes are:

We have removed “stock density” in the title because we have realized that we have emphasized too much the effect of the housing system, and we have added “stock density” in key words.

We have added more information related to other research areas in the introduction.

We have decided to give a visual change to table 5, thus we have added different background colors to make a better compression of the table.

We have modified the discussion for better connection between sections.

Therefore, we would like to thank the time and effort of both the reviewers and the editor that help us improve the manuscript.

Reviewer 2 Report

The authors present and discuss Phenotypic and genetic components for growth, morphology and flesh quality traits of meagre (Argyrosomus regius) reared in tank and sea cage with different stock density. It is helpful to estimate genetic parameters of the main traits that are of fish farming interest, such as growth performance, morphology, and flesh composition, for the first time in meagre, to be considered as a strategy in a selective breeding program. Although there are no fundamental flaws in the manuscript there are some areas that would benefit from further clarification and some question that should be corrected. Suggestions to make major modifications.

  1. The simple Summary needs to be rewritten and should be more simplified.
  2. The result expression part in the abstract is more precise, whether it is a significant decrease or a significant increase.
  3. Introduction, the authors should provide more information on the meagre, especially on the current progress of nutritional research? Since this species is not studied worldwide it would be interesting to enlighten the readers about it.
  4. Since this study is related to flesh quality, and the current experimental data are not enough, it is suggested to supplement important data related to flesh quality such as muscle texture, amino acids and fatty acids, which will greatly improve the MS.
  5. Any information that supports the sequences were fully validated is available(Amplification efficiency)? If so, please provide it in the MS.
  6. Discussion section lack of a brief initial background of the experiment and should be re-written connecting and blending together each section.
  7. References, please update it with more recent works.

Author Response

(The authors gave the same response as above.)

Round 2

Reviewer 1 Report

The authors have provided satisfactory answers to all my comments, and the manuscript was greatly improved. I consider that there are minor flaws in the English speech that should be addressed, maybe by sending the manuscript to a native speaker for revision. Other than that the manuscript is ready to be accepted for publication

Author Response

  • The authors have provided satisfactory answers to all my comments, and the manuscript was greatly improved. I consider that there are minor flaws in the English speech that should be addressed, maybe by sending the manuscript to a native speaker for revision. Other than that the manuscript is ready to be accepted for publication

First of all, thank you for your help in improving this manuscript, as your comments have helped us improve it. At all times, the first, second and this last version have been reviewed by a native speaker specialized in technical language. In this last review we have paid special attention to the language style.

Reviewer 2 Report

  1. Since this study is related to flesh quality, and the current experimental data are not enough, it is suggested to supplement important data related to flesh quality such as muscle texture, amino acids and fatty acids, which will greatly improve the MS.

Author Response

  1. Since this study is related to flesh quality, and the current experimental data are not enough, it is suggested to supplement important data related to flesh quality such as muscle texture, amino acids and fatty acids, which will greatly improve the MS.

It is the first time that genetic parameters for fish morphology traits and flesh composition have been estimated in meagre. Therefore, due to the novelty of our results, and the relevance of these traits for the industry, we consider that the paper should be considered for its publication at this stage. The paper deal with some quality flesh traits but these are not the main part. The paper is more focused on growth traits and the fish morphology. We have added flesh composition because it is important to know how fat percentage can be modified when fish are selected by growth. However, in the end, genetic correlations are estimated without accuracy.

We are aware that we must improve these estimates and analyze more flesh quality traits. As we commented in the previous review, the study of the fatty acid profile in meagre is currently being carried out, but it takes a long time because we need a big amount of data to get accuracy in the estimates. We want these results for a future publication.

We would also like to thank you for your opinion and comments, as they help us to improve this and future manuscripts.
